# New Role for Growth/Differentiation Factor 15 in the Survival of Transplanted Brown Adipose Tissues in Cooperation with Interleukin-6

**DOI:** 10.3390/cells9061365

**Published:** 2020-06-01

**Authors:** Masako Oka, Norihiko Kobayashi, Kazunori Matsumura, Miwako Nishio, Kenta Nakano, Tadashi Okamura, Hitoshi Okochi, Tamiko Minamisawa, Kiyotaka Shiba, Kumiko Saeki

**Affiliations:** 1Department of Disease Control, Research Institute, National Center for Global Health and Medicine, Tokyo 162-8655, Japan; okamako555@gmail.com (M.O.); nrhkkobayashi@gmail.com (N.K.); cc26732@yahoo.co.jp (K.M.); 2Department of Laboratory Molecular Genetics of Hematology, Graduate School of Medical and Dental Sciences, Tokyo Medical and Dental University, Tokyo 113-8510, Japan; mnishio.lmg@tmd.ac.jp; 3Department of Laboratory Animal Medicine, Research Institute, National Center for Global Health and Medicine, Tokyo 162-8655, Japan; kennakano@ri.ncgm.go.jp (K.N.); okamurat@ri.ncgm.go.jp (T.O.); 4Department of Infectious Diseases, Section of Animal Models, Research Institute, National Center for Global Health and Medicine, Tokyo 162-8655, Japan; 5Department of Regenerative Medicine, Research Institute, National Center for Global Health and Medicine, Tokyo 162-8655, Japan; hokochi@ri.ncgm.go.jp; 6Division of Protein Engineering, Cancer Institute, Japanese Foundation for Cancer Research, Tokyo 135-8550, Japan; tminamisawa@jfcr.or.jp (T.M.); kshiba@jfcr.or.jp (K.S.); 7ID Pharma Co., Ltd., Ibaraki 300-2611, Japan; 8I’rom Group Co., Ltd., Tokyo 102-0071, Japan

**Keywords:** GDF15, IL6, brown adipocyte, human pluripotent stem cells, intraperitoneal transplantation

## Abstract

To identify factors involved in the earliest phase of the differentiation of human embryonic stem cells (hESCs) into brown adipocytes (BAs), we performed multi-time point microarray analyses. We found that growth/differentiation factor 15 (GDF15) expressions were specifically upregulated within three days of differentiation, when expressions of immature hESC markers were sustained. Although GDF15 expressions continued to increase in the subsequent differentiation phases, *GDF15*-deficient hESCs differentiated into mature BAs (Day 10) without apparent abnormalities. In addition, *GDF15*-deficient mice had normal brown adipose tissue (BAT) and were metabolically healthy. Unexpectedly, we found that interleukin-6 (IL6) expression was significantly lowered in the BAT of *GDF15*^-/-^ mice. In addition, *GDF15*^-/-^ hESCs showed abortive IL6 expressions in the later phase (>Day 6) of the differentiation. Interestingly, GDF15 expression was markedly repressed throughout the whole course of the differentiation of *IL6^-^*^/-^ hESCs into BAs, indicating IL6 is essential for the induction of GDF15 in the differentiation of hESCs. Finally, intraperitoneally transplanted BAT grafts of *GDF15*^-/-^ donor mice, but not those of wild-type (WT) mice, failed in the long-term survival (12 weeks) in *GDF15*^-/-^ recipient mice. Collectively, GDF15 is required for long-term survival of BAT grafts by creating a mutual gene induction loop with IL6.

## 1. Introduction

Obesity is one of the major risk factors for lifestyle diseases such as cardiovascular diseases (CVDs), diabetes, and cancers. However, there is a puzzling phenomenon known as “obesity paradox”, where patients with CVD show a better prognosis if classified as overweight or class I obese [1]. Overweight/obesity imposes a high exercise load on muscles and bones in daily life. Accordingly, overweight and class I obese patients, as long as they are aiming to get adequate exercise in their everyday lives, may behave better owing to the relatively greater lean mass compared to normal or underweight patients. Therefore, the ideal body weight under a specific pathological condition may vary depending on the case, although obesity per se is unpreferable for the maintenance of good health. There are many weight-loss medicines, all of which have concerns especially in view of genotoxicity. Bariatric surgeries are considered as the last resort; however, there are serious side effects. Some patients are compelled to use opioids chronically and are even at risk of attempting suicide [2]. Therefore, it is preferable to take a more natural approach to the maintenance of body weight. From this standpoint, brown adipose tissue (BAT), which exerts a high thermogenic activity and is therefore commonly known as “a burning fat” or “a slenderizing fat”, has been attracting increasing attention as a target for the development of weight reduction methods.

BAT contributes not only to the maintenance of body temperature under cold environments but also to the improvement of metabolism by secreting various BATokines [3,4,5]. Although small-sized mammals have abundant BATs to compensate for high heat loss per body surface, large-sized mammals including humans have functional BATs as well [6,7,8,9,10]. BATs locate in the specific sites of the body including the interscapular space, where the largest BAT locates. For unknown reasons and by unknown mechanisms, interscapular BAT (iBAT) disappears during early infancy in humans. In adult humans, functional BATs remain in specific regions such as supraclavicular, deep neck, axillar, and paravertebral areas. There is an inverse correlation between the amounts of BAT, which can be measured using ^18^F-FDG-PET, the degree of obesity [11,12,13] and serum glucose levels [14]. Moreover, it has been reported that increments in BAT amounts by cold acclimation improved insulin sensitivity [15]. Therefore, BAT can provide an ideal material for the transplantation therapy for morbid obesity with disordered glucose metabolism. However, there are two problems. One concerns the stable provision of human BAs as a transplantation material. Since there are multiple hurdles in obtaining high-quality human BA samples from a living body from technical, economic, and ethical standpoints [16], other measures should be taken. We previously established a method for a directed differentiation of human pluripotent stem cells into functional BAs without gene transfer [17,18]. Although this technique enabled a stable provision of high-quality human BAs, the differentiation medium contained a high concentration of hematopoietic cytokine cocktail. Hence, there was a risk of adverse effects due to contamination by residual hematopoietic cytokines. To address this issue, we recently upgraded the method so that the differentiation is executed without using synthetic cytokines [19]. Thus, our upgraded gene-transfer-free and cytokine-free differentiation technique can provide an ideal material for transplantation therapy.

The other problem concerns the route of transplantation. Murine transplantation experiments have shown that subcutaneous transplanted BAT exerted only short-term (~3 weeks) effects, whereas intraperitoneally transplanted BAT exerted long-term (>16 weeks) effects [20]. The latter route, however, has a high risk of side reactions such as peritonitis. To address this issue, the molecular basis for long-term survival of intraperitoneally transplanted BAT grafts must be clarified. It is known that BAT graft-derived IL6 is required for anti-obesity effects [20]. It is also known that BAT graft-derived IL6 augmented IL6 mRNA expressions in the BATs of recipient mice [18]. These findings suggest that IL6 creates a feed-forward loop to strengthen functional integrality of BATs, which are distributed in specific sites in the body. Nevertheless, if recombinant IL6 is subcutaneously injected, it might cause adverse effects due to its proinflammatory activities.

During embryogenesis, BAs are differentiated from myf5-positive myoblasts [21]. Since the in vivo differentiation process is properly reproduced in our method for the differentiation of human pluripotent stem cells into BAs [17,18], our system is applicable to the study to discover still unidentified factors that are working during the differentiation into BAs. In our system, it takes about 10 days for immature human pluripotent stem cells to be differentiated into mature BAs. The myf5-positive myoblast stage corresponds to Day 6 in our differentiation system [17]. Because the molecular events that are working in earlier phases than the myoblast stage remain elusive, we focused our study on the earliest phase of the differentiation (<Day 3). Although the expressions of immature human embryonic stem cell (hESC) marker genes remained at high levels during this early phase, cells have already been committed to differentiation since Day 3 hESCs cannot recover pluripotency. Therefore, certain crucial molecular events should have occurred before Day 3. To identify such events, we performed multi-time point microarray analyses after 3–72 h from an induction of differentiation.

Here we show that the expression of growth/differentiation factor 15 (GDF15) is induced in this early phase of the differentiation. GDF15 is a pleiotropic cytokine and exerts multiple physiological or pathological effects such as inflammatory responses, metabolism regulation, and tumorigenesis. In relation to IL6, a report shows that *GDF15*^-/-^ mice have a decreased percentage of inflammatory IL6-positive leukocytes in atherosclerotic vessels [22], suggesting a link between GDF15 and IL6 in gene inductions. In the current study, we show that GDF15 and IL6 gene expressions are mutually induced in the course of a directed differentiation of human embryonic stem cells (hESCs) into BAs. We further show that, although GDF15 is dispensable for the development of murine BAT in vivo and the differentiation of hESCs into BAs in vitro, GDF15 plays indispensable roles in the survival of transplanted BAT grafts. The role of GDF15 in neo-angiogenesis and the mode of GDF15 secretion will also be discussed.

## 2. Materials and Methods

### 2.1. Cells and Reagents

KhES-3, a human embryonic stem cell (hESC) line, was provided by the Institute for Frontier Life and Medical Sciences, Kyoto University, Kyoto, Japan [23] and maintained by a feeder-free system as previously reported [19]. Differentiation of hESCs into BAs was performed by applying either an original method [17,18] or an upgraded cytokine-free method [19]. The genome-edited sublines of hESCs were established as previously described [24] using an Alt-R^®^ CRISPR/Cas9 genome editing system (Integrated DNA Technologies, Inc. Coralville, IA, USA), in which Alt-R^®^ Recombinant Cas9 nuclease and Alt-R^®^ tracrRNA were included. Briefly, crRNAs were synthesized by using guide RNAs according to the manufacturer’s instructions. For the establishment of GDF15^-/-^ hESC sublines, two guide RNAs, whose nucleotide sequences were 5′-GGGCGGCCCGAGAGATACGCAGG-3′ and 5′-TCCACTGTGCACCTGCGCGGAGG-3′, were used to delete the exon 2 of the GDF15 gene. For the establishment of *IL6*^-/-^ hESC sublines, two guide RNAs, whose nucleotide sequences were 5′-GGAGAAGGCAACTGGACCGAAGG-3′ and 5′-TTTGTCAATTCGTTCTGAAGAGG-3′, were used to delete the exon 2 of the IL6 gene. Each crRNA was annealed with tracrRNA to prepare the crRNA:tracrRNA complex. Immediately before transfection, the crRNA:tracrRNA:Cas9 complex was prepared using 1 µL crRNA:tracrRNA complex, 1 µL Alt-R^®^ Recombinant Cas9 nuclease, 1.5 µL 1 M KCl, 1 µL 200 mM HEPES buffer (pH 7.4), and 5.5 µL water. The crRNA:tracrRNA:Cas9 complex was added to 8 × 10^5^ hESC suspension in the Nucleofector human stem cell kit ver2 (Lonza Group AG, Basel, Switzerland) and transfection was executed by using Nucleofector I^®^ (Lonza Group AG) with the program #B16. Then, the transfected hESCs were seeded into an iMatrix-coated 60 mm dish for the successive cloning procedure. The PCR fragments that cover the targeted genomic region were subjected to Sanger sequencing (GENEWIZ Japan Corp., Saitama, Japan) to select properly gene-edited clones. MIN6 cells (clone 4) [25,26] were maintained and used for insulin secretion assay using the supernatant of hESC-derived BAs as previously described [19].

### 2.2. Multi-Time Point Microarray Analyses

hESCs were differentiated into BAs by applying an original method. Cells were harvested after 3, 4, 6, 8, 10, 12, 14, 16, 20, 24, 28, 32, 36, 42, 48, 54, 60, 66, and 72 h from the induction of the differentiation and total RNAs were isolated using TRIzoL^®^ Reagent (Thermo Fisher Scientific Inc., Waltham, MA, USA). Microarray analyses were performed by Oncomics Co., Ltd. (Nagoya, Japan) using SurePrint G3 Human Gene Expression 8 × 60 K v2 (Agilent Technologies, Santa Clara, CA, USA). The results were uploaded to GEO profiles in the National Center for Biotechnology Information (NCBI) with the accession number GSE114097.

### 2.3. Quantitative Reverse Transcription Polymerase Chain Reaction (qRT-PCR)

Total RNA was extracted from (1.5–2.0) × 10^6^ cells using an RNeasy Mini Kit (QIAGEN, Hilden, Germany) along with DNase I treatment according to the manufacturer’s instructions. cDNA was prepared from 2.5 µg RNA via RT reaction in a 20 µL solution using SuperScript IV VILO (Thermo Fisher Scientific Inc., Waltham, MA, USA). qPCR was performed by applying 2 µL of 1/10-diluted cDNA template StepOnePlus^TM^ real time PCR System (Thermo Fisher Scientific Inc., Waltham, MA, USA). The nucleotide sequences of the primers used in qPCR were as follows: PAI-1, forward: TCTGAGAACTTCAGGATGCAGAT, reverse: CCACGTAGGATGGGGGATG; GDF15, forward: ATACTCACGCCAGAAGTGCGG, reverse: CTTGCAAGGCTGAGCTGACG; PRDM16, forward: CGAGGCCCCTGTCTACATTC, reverse: GCTCCCATCCGAAGTCTGTC; PPARG, forward: AGCCTGCGAAAGCCTTTTGGTG, reverse: GGCTTCACATTCAGCAAACCTGG; IL6, forward: ACTCACCTCTTCAGAACGAATTG, reverse: CCATCTTTGGAAGGTTCAGGTTG; and GAPDH, forward: CCACTCCTCCACCTTTGAC, reverse: ACCCTGTTGCTGTAGCCA. The results for each mRNA level were normalized against those for GAPDH.

### 2.4. Preparation of Extracellular Vesicles (EVs)

The supernatant (90 mL) of human ESC-derived BAs (BA-SUP) was subjected to sequential centrifugations at 300× *g* for 10 min (Model 5500 Centrifuge, KUBOTA Corp., Tokyo, Japan), at 2000× *g* for 10 min (Model 5500, KUBOTA Corp.), at 10,000× *g* for 30 min (L-90K with SW32Ti rotor and Ultra-Clear™ Centrifuge tubes (344058), Beckman Coulter LLC, Brea, CA, USA), and 160,000× *g* for 70 min (L-90K with SW32Ti rotor and Ultra-Clear™ Centrifuge tubes (344058), Beckman Coulter LLC, Brea, CA, USA). In each step, the pellets were washed once with 30 mL of PBS. All centrifugations were done at 4 °C.

### 2.5. Immunostaining and Western Blotting

Immunostaining was performed as previously described [19]. The 1st antibody reaction was performed by using either a rabbit polyclonal anti-human GDF15 antibody (sc66904; Santa Cruz Biotechnology Inc., Santa Cruz, CA, USA) or a normal IgG (sc-2027; Santa Cruz Biotechnology, Dallas, TX, USA), and the 2nd antibody reaction was performed by using an Alexa Fluor^®^ 594-conjugated goat anti-rabbit IgG antibody (A11036; Thermo Fisher Scientific Inc., Waltham, MA, USA). The samples were observed either under a BZ-X710 All-in-One fluorescence microscope (KEYENCE Corp., Osaka, Japan). Western blotting was performed according to the method previously described [19] using the following antibodies in the 1st antibody reactions: a rat polyclonal anti-human GDF15 antibody (sc66904; Santa Cruz Biotechnology Inc., Santa Cruz, CA, USA), a mouse monoclonal anti-human SMAD2 antibody (3103S; Cell Signaling Technology Inc., Danvers, MA, USA), a rabbit polyclonal anti-phosphorylated SMAD2 antibody (3101S; Cell Signaling Technology Inc., Danvers, MA, USA), a mouse monoclonal anti-human CD9 antibody (SHI-EXO-M01) (SHIONOGI & CO., LTD., Osaka, Japan), a mouse monoclonal anti-human CD81 (11-558-C100, EXBIO Praha, a.s., Vestec, Czech Republic), a mouse monoclonal anti-ATP5A (ab14748, Abcam plc, Cambridge, UK), a rabbit polyclonal anti-human Hsp70 (EXOAB-Hsp70A-1, System Biosciences, LLC, Palo Alto, CA, USA), an anti-tubulin beta antibody (ab134185; Abcam plc), and a mouse monoclonal anti-human β-actin (A1978, Sigma-Aldrich Co. LLC, St. Louis, MO, USA).

### 2.6. Animal Experiments

For establishing GDF15 knockout mice, a CRISPR/Cas9-mediated genome editing system was applied as described previously [27] with modifications. Briefly, an expression vector for a single-guide (sg) RNA, whose nucleotide sequence was 5′-GCTGCTACTCCGCGTCAACCGGG-3′ and 5′-TATGATGACCTGGTGGCCCGGGG-3′, with a T7 promoter was synthesized, and transcribed in vitro using a MEGAshortscript Kit (Life Technologies, Carlsbad, CA, USA). hCas9 mRNA was synthesized using an mMESSAGE mMACHINE T7 kit (Life Technologies) and was polyadenylated with a polyA tailing kit (Life Technologies). Introduction of purified sgRNA and hCas9 mRNAs into fertilized eggs from C57BL/6NCr (Japan SLC Inc., Hamamatsu, Japan) by electroporation was performed as described previously [28]. After the electroporated oocytes were cultured overnight in vitro, two-cell embryos were transferred into the oviducts of pseudopregnant ICR females (CLEA Japan Inc., Tokyo, Japan). Tail genomic DNA of offspring was isolated using standard methods. After checking the DNA sequence of the GDF15 genome of the offspring, we chose a male mouse, where exon 2 of the GDF15 gene was completely deleted (Appendix A), as a mating partner of wild-type (WT) female mice. Offspring were subjected to genomic PCR analyses using a forward primer (5′-caaatccgctagggttgtgt-3′) and a reverse primer (5′-ccccctaattctgggatgtt-3′). There primers amplify a 1760 bp fragment, which corresponds to the normal GDF15 gene-derived fragment, and an 887 bp fragment, which corresponds to the edited GDF15 gene-derived fragment lacking a part of intron 1, whole exon 2, and a part of intron 2. F1 mice were further crossed for generation of GDF15 knockout mice.

For evaluating glucose metabolism, 5-week-old mice were fed a high fat diet (D12492, Rodent Diet With 60 kcal% Fat, Research Diets, Inc., New Brunswick, NJ, USA) for 8 weeks. Then, an oral glucose tolerance test (OGTT) was performed by orally administrating 0.1 g/mL glucose after fasting for 16 h as previously reported [17].

For BAT transplantation, interscapular BAT were removed and intraperitoneally transplanted to syngeneic recipients according to the method presented by Stanford et al. [20]. After 12 weeks following transplantation, the transfected iBAT grafts were collected and tissues were macroscopically examined.

All mice were housed in air-conditioned animal rooms at an ambient air temperature of 22 ± 2 °C and relative humidity of 50% ± 15%, under specific pathogen-free conditions with a 12-h light/dark cycle (8:00 to 20:00 light period). They were fed a standard diet (CE-2, CLEA Japan, Inc., Tokyo, Japan) with free access to drinking water. All animal experiments were approved by the Animal Care and Use Committee of the National Center for Global Health and Medicine (NCGM) Research Institute (Permission Numbers 19040, 19041, 19048, and 19049) and conducted in accordance with institutional procedures.

### 2.7. Statistics

Experiments were performed in multiplicate both in vitro (*n* = 3 or 4 mice) and in vivo (*n* = 8–15 mice) as comparison between two groups and the data were analyzed by *t*-test. All values are expressed as mean ± standard deviation (SD).

## 3. Results

### 3.1. Gene Expression Dynamics in the Earliest Phase of the Differentiation of hESCs

To investigate the gene expression profile in the earliest phase of the differentiation of hESCs into BAs, we performed multi-time point microarray analyses using cell samples that were harvested from 3 h to 72 h after the induction of the differentiation. During this early phase, expressions of immature hESC markers such as NANOG, SOX2, and POU5F1 were maintained at high levels (Figure 1a). The expression level of KLF4, one of the Yamanaka factors [29] that induces initialization of somatic cells, slightly increased (Figure 1a, right). Because the cells had already undergone differentiation at this time, there should have been genes whose expressions showed obvious changes even in this earliest phase. We found that expressions of LEFTY1, LEFTY2, and NODAL, which are involved in axis determinations in early embryogenesis, clearly showed time-dependent decrements (Figure 1b). Interestingly, the expression of a beige cell marker, CITED1, which can discriminate beige cells from BAs [30], showed a similar reduction although its expression levels were significantly lower than those of LEFTY1, LEFTY2, and NODAL (Figure 1b, right). Developmentally, beige cells and BAs are derived from distinct cell lineages, that is, beige cells are derived from pericytes [31,32], whereas BAs are derived from MYF5-positive myoblasts [21]. Therefore, the time-dependent decrements in LEFTY1, LEFTY2, NODAL, and CITED1 reflected the fact that the cells had already been prepared for a direct differentiation into BAs.

Thus, despite the sustained expressions of immature hESC marker genes, a highly organized gene expression-controlling system was already working for the cells to undergo a directed differentiation into BAs.

### 3.2. Induction of GDF15 during the Differentiation of hESCs into BAs

We hypothesized that there must be genes that showed increasing expressions during this early phase of differentiation (3–72 h) because the waning of “non-BA” gene expressions (Figure 1b) should not be sufficient for an induction of a directed differentiation of hESCs into BAs. To evaluate this hypothesis, we checked the microarray data and found that twenty-two genes showed increasing expressions over time (Appendix A). Among them, we focused our study on GDF15 for the following two reasons. First, GDF15 showed the most remarkable fold increments with highest signal intensities (Appendix A). Second, GDF15 belongs to a BMP/GDF family of the TGFB superfamily [33], which is known to play important roles in morphogenesis and tissue differentiation. Among TGFB superfamily genes, only twelve genes showed positive signals in microarray analyses (Figure 2a). Interestingly, GDF15 was the only TGFB superfamily gene that showed increasing expressions in this phase (Figure 2a, red line). Western blotting studies confirmed the induction of GDF15 protein expression (Figure 2b, upper), the phosphorylation of its downstream transcription factor SMAD2 (Figure 2b, upper), and the inductions of the mRNA (Figure 2c) and protein (Figure 2d) of PAI, a downstream target gene of activated SMAD2. Moreover, the expressions of GDF15 mRNA continued to increase in the successive phases (Day 3 to Day 8) (Figure 2e). Immunostaining studies further confirmed the expression of GDF15 protein in mature BA samples at Day 10 (Figure 2f).

Therefore, GDF15 expression is induced in the course of the differentiation of hESCs into BAs from the earliest phase (3–72 h) to the terminally differentiated phase (Day 10).

### 3.3. GDF15 Is Unessential for the Development of Murine BAT and the Differentiation of hESCs into BAs

Since GDF15 is involved in the improvement of metabolism [34,35,36,37,38] via an activation of BAT, at least in part [39], we generated GDF15 knockout mice to evaluate the impact of GDF15 deficiency on the development of BAT (Appendix A). Although the expression levels of GDF15 were considerably low in iBAT, we could confirm that GDF15 expression was eliminated in the knockout mice (Figure 3a, Appendix A). Contrary to our expectation, the lack of GDF15 expression did not affect the morphology of iBAT (Figure 3b). Moreover, there were no significant differences in body weight (Figure 3c) or glucose tolerance (Figure 3d) between *GDF15*^-/-^ and wild-type mice. Therefore, GDF15 is not essential for the development or functional maturation of BAT, at least in mice.

We next investigated the role of GDF15 in the differentiation of human BAs. For this purpose, we established *GDF15*^-/-^ hESCs by using a CRISPR/Cas9 system and subjected them to the differentiation into BAs. We found no morphological abnormalities in *GDF15*^-/-^ hESC-derived BAs. Moreover, the expressions of *PRDM16* and *PPARG*, which are the two crucial transcription factors for the induction of the differentiation into BAs, were appropriately induced in *GDF15*^-/-^ hESCs (Figure 4a). We also evaluated the metabolism-improving activity of *GDF15^-^*^/-^ hESC-derived BAs. Since the supernatant of hESC-derived BAs (BA-SUP) has an ability to augment basal insulin secretions by pancreatic beta cells [19], we compared the insulin secretion-enhancing potential of the BA-SUP. Again, we found no differences in the activities of the BA-SUP to enhance insulin secretion between *GDF15*^-/-^ and wild-type (WT) hESCs (Figure 4b), indicating that GDF15 is dispensable for both the differentiation and functional maturation of BAs.

Therefore, despite a remarkable induction of GDF15 in the course of the differentiation of hESCs into BAs, GDF15 is dispensable for both the development of murine BAT and the differentiation of human BAs.

### 3.4. Mutual Gene Inductions between GDF15 and IL6 during the Differentitaion of BAs 

To understand the role of GDF15, we examined the expression profile of *GDF15* using an open database BioGPS [40] and found that the tissue distribution pattern of *GDF15* closely resembled that of *IL6* (Appendix A). IL6 is recognized as a proinflammatory cytokine; nevertheless, it also serves as a BATokine [3,4,5]. In addition, IL6 is required for intraperitoneally BAT graft to exert metabolism-improving effects [20]. As mentioned in the previous section, GDF15 exerts metabolism-enhancing activities when overexpressed [34,35]. Therefore, it seems that GDF15 and IL6 share high similarities from functional and distributional standpoints. Moreover, the fact that BAT-graft-derived IL6 augmented the expression of IL6 in the recipient BATs [20] suggests that there is function linkage among various BATs in the body.

Based on these findings, we hypothesized that GDF15 was involved in the induction of IL6 in BAs. To validate this hypothesis, we assessed the effects of GDF15 deficiency on the expression of IL6 in murine iBATs. We found that IL6 expressions were considerably lowered in the iBAT of GDF15^-/-^ mice (Figure 5a). By contrast, the expression levels of IL6 in the liver and muscles showed no tendencies by the absence of GDF15 (Appendix A). To evaluate the hypothesis in human cases, we examined IL6 expressions in *GDF15^-^*^/-^ hESCs in the course of the differentiation into BAs. We found that IL6 induction became abortive after Day 6, which corresponds to the phase after the myoblast stage [17] (Figure 5b). Therefore, GDF15 is required for a sustained induction of IL6 in phases later than the myoblast stage.

We also evaluated the impact of IL6 on the induction of GDF15. For this purpose, we established *IL6^-^*^/-^ hESCs by using a CRISPR/Cas9 system and examined the expressions of *GDF15* in the course of the differentiation into BAs. We found that GDF15 inductions were blunted in all three independent clones of *IL6*^-/-^ hESCs (Figure 6), indicating that IL6 plays an important role in inducing GDF15 in the differentiation of hESCs into BAs.

Collectively, IL6 and GDF15 together create a feed-forward loop for gene inductions in the course of the differentiation of hESCs into BAs.

### 3.5. GDF15 is Required for Long-Term Survival of Intraperitoneally Transplanted BAT Grafts in Mice

Although IL6 is dispensable for the development of BAT, it is essential for the transplanted BAT grafts in order to exert long-term metabolism-improving effects [20]. Since GDF15 is required for the sustained expression of IL6, we hypothesized that the absence of GDF15 might hamper the long-term survival of the transplanted BAT grafts. To evaluate this hypothesis, iBATs of WT or those of *GDF15*^-/-^ mice were intraperitoneally transplanted into *GDF15*^-/-^ mice and, after 12 weeks, the morphologies of the BAT grafts were macroscopically evaluated. We found that iBAT grafts of *GDF15*^-/-^ mice were shrunken and no angiogenesis was detected around the grafts (upper), whereas those of WT mice retained their sizes and showed a reddish appearance as a result of neo-angiogenesis in the soft tissue around the grafts (lower). Similar results were obtained when iBAT grafts were subcutaneously transplanted. Consistently with a report that subcutaneous transplantation of iBAT failed in exerting long-term effects on metabolism improvement [17,20], subcutaneously transplanted iBAT showed a shrunken appearance with inadequate angiogenesis. Nevertheless, we could detect its existence after 12 weeks from transplantation (Appendix A). By contrast, subcutaneously transplanted iBAT of *GDF15*^-/-^ mice had disappeared (Appendix A).

When we performed histological analyses, we found an intriguing result. In the case of GDF15^-/-^ mice-derived BAT grafts, parenchymal cells exclusively underwent death via a lytic (necrotic) or condensed (apoptotic) process. Reflecting the shrinkage of parenchymal areas, interlobular spaces were significantly widened. Surprisingly, the basic framework of arteries was highly maintained. For example, the structures of elastic membranes showed a healthy appearance (Figure 7b, left, arrows). There was no infiltration of inflammatory cells, which is compatible with the fact that there was no neo-angiogenesis around *GDF15*^-/-^ mice-derived BAT grafts (Figure 7a, upper). On the other hand, parenchymal cells of the WT mice-derived BAT grafts showed a healthy appearance (Figure 7b, right).

Therefore, GDF15 is required for long-term survival of BAT grafts despite its dispensability for the development of BAT per se.

### 3.6. Mode of GDF15 Secretion from hESC-Derived BAs

To obtain mechanistic insights into the involvement of GDF15 in the survival of BAT grafts, we examined the mode of GDF15 secretion. Since some cytokines, such as Wnts, are known to be secreted as a component of exosomes [41], we examined whether GDF15 was secreted as a component of extracellular vesicles (EVs) or released in a soluble form. For this purpose, the culture supernatant of hESC-derived BAs was fractionated by a sequential centrifugation (300× *g*, 2000× *g*, 10,000× *g*, and 160,000× *g*). Each precipitated fraction was subjected to Western blotting using antibodies against GDF15, canonical EV markers (CD9 and CD81), a mitochondrial marker (ATP5A), and ubiquitously expressed proteins (HSP70 and β-actin). Surprisingly, GDF15 existed in the fractions that contained larger EVs. GDF15 protein was detected in 300× *g*- and 2000× *g*-precipitated fractions, which include the mitochondria marker, ATP5A, but not in the 10,000× *g*-precipitated fraction, which may include microvesicles, or in the 160,000× *g*-precipitated fraction, which includes exosomes (Figure 8).

Therefore, GDF15 may be secreted as a component of large-sized EVs, at least in the case of hESC-derived BAs.

## 4. Discussion

In the present study, we showed the following: (1) GDF15 is induced at the earliest phase of differentiation and its expression continues to be upregulated in later phases; (2) GDF15 is dispensable for BAT development but required for long-term survival of BAT grafts; (3) there is a dynamic gene induction system between GDF15 and IL6, namely IL6 induces GDF15 expressions before the myoblast phase and GDF15 and IL6 mutually induce their expressions in later phases; and (4) GDF15 exists in the fractions that contain larger EVs in the supernatant of hESC-derived BAs. The mutual gene-inducing system between GDF15 and IL6 along with the accumulation of GDF15-containing larger-sized EVs around BAT grafts may be advantageous for establishing a stable blood supply system via IL6-mediated neo-angiogenesis, which contributes to long-term survival of BAT grafts (Figure 9). Since GDF15 is unessential for IL6 induction in the liver and skeletal muscles (Appendix A), the requirement for GDF15 in the survival of grafts seems to be a selective phenomenon to BAT. Whether the effect of GDF15 on IL-6 expression and that of IL6 on GDF15 expression are direct or indirect remains elusive. Genomic information from the UCSC Genome Browser (https://genome.ucsc.edu/) indicates that there is a relatively large (~300 bp) STAT3-binding area within the enhancer region of the GDF15 gene as shown by transcription factor CHIP-seq analyses, whereas there are no SMAD-binding sites shown within the enhancer region of the IL6 gene. Therefore, IL6 may possibly induce GDF15 gene expression directly via STAT3 activation, whereas GDF15 acts indirectly to induce the IL6 gene. Further investigation is required for the elucidation of the molecular mechanism of mutual gene induction between IL6 and GDF15.

Our findings give one possible solution to overcome the problem regarding the route of BAT transplantation—subcutaneous transplantation is safer but disadvantageous from the standpoint of graft survival, whereas intraperitoneal transplantation is advantageous for graft survival but has a high risk of complications. In intraperitoneal transplantation, BAT grafts were confined in a narrow space on the pelvic floor, being covered by intestines. Therefore, GDF15-containing larger-sized EVs may exist at high concentrations in peri-graft spaces without disseminating throughout the abdominal cavity. If there is a feasible technique to sequestrate GDF15-containing EVs in narrow spaces around the subcutaneously transplanted grafts, it may lengthen the survival time of the grafts.

The unique histological change of the transplanted BAT grafts of *GDF15^-^*^/-^ mice indicates that the basic framework of BAT is resistant to ischemia-dependent degeneration. This seems to be an unusual case regarding the fate of ischemic grafts. BAT expresses high levels of uncoupling protein 1 (Ucp1), which is known to reduce the production of reactive oxygen species (ROS) by mitochondria. Therefore, BAT may be protected from ROS-mediated damages under ischemic conditions. Interestingly, even in the shrunken *GDF15*^-/-^ mice-derived BAT grafts, immunoreactivity of Ucp1 protein was detected (Appendix A). Accordingly, it may be possible that Ucp1 protein served as a blocker against ROS production in the transplanted grafts to save the tissue framework from destruction, although parenchymal cells per se had undergone death as a result of nutrient deficiency due to a lack of blood supply. Although it remains elusive when massive cell death in *GDF15*^-/-^ mice-derived BAT grafts started, it seems that the commitment of cell death occurred at an early phase after transplantation. We observed in our previous experiments on the organ culture of iBAT that the morphology of the BAs turned to a white adipocyte-like feature, with a large monolocular lipid droplet, within 24 h (unpublished observations). On the other hand, dead BAs in *GDF15*^-/-^ mice-derived BAT grafts still bore multilocular lipid droplets. Therefore, signals to induce cell death should have been transmitted within 24 h after transplantation. Future investigation may clarify the detailed process of the unique cell death executed in ischemic BAT.

Various roles for GDF15 to play in metabolism regulation have been shown. For example, GDF15 is involved in the control of food intake [34,35,36,37,38], reduction of body weight [34,35,36,37,38,39], improvement of glucose metabolism [34,35,36,37,38,39], upregulation of energy expenditure via activating thermogenesis in BAT and lipolysis in white adipose tissue (WAT) [39], and enhancement of mitochondrial activity in skeletal muscles [38]. Roles for GDF15 in tumorigenesis have also been shown. For example, GDF15 is involved in the progression of esophageal squamous cell carcinomas [42], epithelial–mesenchymal transition (EMT) and metastasis in colorectal cancers [43], and an enhancement of the tumor-initiating and self-renewal potential of multiple myeloma cells [44]. It was also reported that enrichment of BA characteristics was observed in the xenografts of breast cancer cell lines in mice [45]. Although causal association was not determined, BA-derived GDF15 might possibly contribute to the progression of breast cancers. Regarding tumorigenesis, however, the role of GDF15 seems rather complicated. Although GDF-15 is overexpressed in a variety of tumors, there are a number of reports showing its anti-tumorigenic effects [39], also reviewed in [46]. Moreover, the involvement of p53, a tumor-suppressor gene, in the induction of GDF15 varies depending on the case [46]. Therefore, it may not be easy to determine the precise role of GDF15 in each event. Regarding the regulation of metabolism, contradictory findings have been reported. Some groups reported that GDF15^-/-^ mice showed higher body weight [36,37] and impaired glucose metabolism [37], whereas other groups, including our group, showed that there were no significant differences in body weight or glucose tolerance between *GDF15^-^*^/-^ and WT mice [47]. Nevertheless, the question may not be so simple. Actually, we did observe severe obesity, diabetes, and serious fatty liver in high-fat diet (HFD)-fed *GDF15*^-/-^ mice when we first obtained three offspring. For unknown reasons, the degree of obesity of our *GDF15^-^*^/-^ line became milder as the generation number became higher. Finally, we were not able to detect any significant differences in body weight or glucose metabolism between *GDF15*^-/-^ and WT mice when we had prepared sufficient numbers of the offspring to perform statistically analyzable experiments. Although we examined breeding conditions, feeding conditions, and seasonal effects, we could not find the reason for the difference in the results. There may be an unexpectedly high redundancy in the signaling events in which GDF15 is involved, and therefore, the phenotypes of *GDF15*^-/-^ mice would substantially vary depending on the case. Alternatively, differences in the mode of the existence of the GDF15 protein (i.e., its presence in the fraction of large-sized EVs, in the fraction of small-sized EVs, or in the soluble fraction) may influence the phenotype of the mice. Future studies will solve the mystery of the hypervariation regarding the functions and effects of GDF15.

Although we could not determine the role of GDF15 under natural conditions, we were able to demonstrate its role in an artificial situation (i.e., survival of BAT grafts). Our current study will be of use when hESC/hiPSC-derived BAs are applied to transplantation therapy for the treatment of morbid obesity.

## 5. Conclusions

GDF15 creates a mutually inducing loop with IL6 in brown adipocytes and, although it is dispensable for BAT development, it plays crucial roles in the long-term survival of intraperitoneally transplanted BAT grafts.

## 6. Patents

The original method for a directed differentiation of human pluripotent stem cells into brown adipocytes is internationally patented (Japan 5998405; United States 9492485; Australia 2012248333; China ZL201280020534.X; EU granted it on 12 June 2018, currently under registration procedure in Switzerland, Germany, France, and the United Kingdom).

A method for preparing a biologically active culture supernatant from brown adipocytes generated from human pluripotent stem cells under a cytokine-free condition is currently Japanese patent pending (Tokugan 2018-152727, Tokugan 2019-026463).

A method of purification and the identification of the molecule with basal insulin secretion-stimulating activity is currently Japanese patent pending (Tokugan 2019-12945).

## Figures and Tables

**Figure 1 cells-09-01365-f001:**
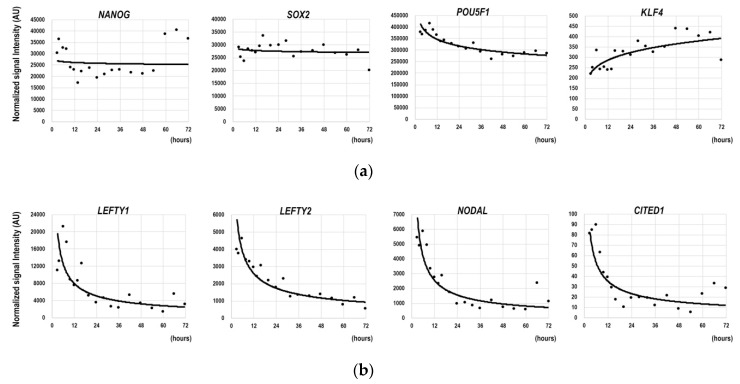
Changes in the gene expression profile in the earliest phase of differentiation. hESCs (KhES-3) were subjected to differentiation into brown adipocytes (BAs). During the time period from 3 h to 72 h after the induction of differentiation, cells were harvested at 19 time points in total and microarray analyses were performed. Horizontal and longitudinal axes indicate hours from the start of differentiation induction and normalized signal values, respectively. Data regarding (**a**) immature pluripotent stem cell marker genes and (**b**) the genes with time-dependent reductions are shown. The power approximate curve is shown on each graph.

**Figure 2 cells-09-01365-f002:**
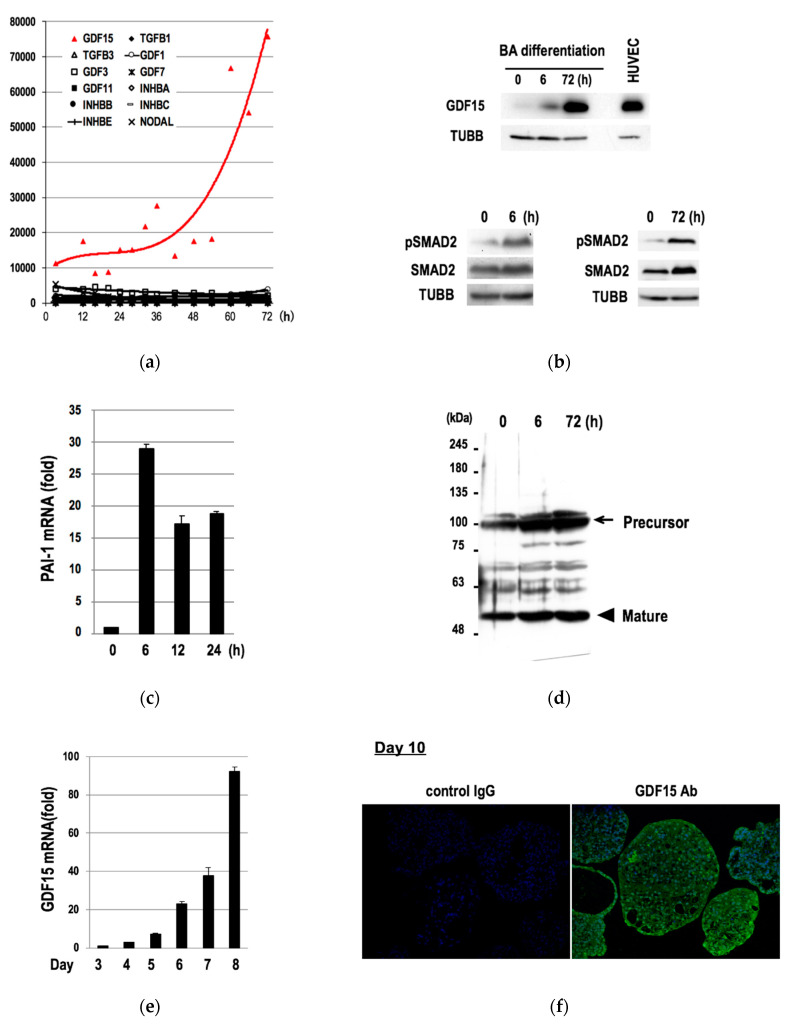
Induction of *GDF15* and an activation of its downstream signaling molecule during the differentiation of hESCs into BAs. (**a**) Results of microarray analyses regarding *TGFB/BMP/GDF* family are shown. The vertical and horizontal axes indicate the signal intensity and the time duration after an induction of differentiation, respectively. The cubic polynomial approximation curve is shown on the graph. (**b**) Western blotting studies for GDF15 protein and beta-tubulin (TUBB) proteins using the differentiating hESCs or human umbilical vein endothelial cells (HUVEC) as a positive control (**upper**), Western blotting for phosphorylated SMAD2 (pSMAD2), total SMAD2, and TUBB proteins at indicated time points (**lower**) are shown. (**c**) qRT-PCR for PAI mRNA. The vertical axis indicates “fold increments” when compared to the data of the samples at 0 h. Data are shown as mean ± SD (*n* = 3). (**d**) Western blotting for PAI precursor protein and mature PAI protein. (**e**) qRT-PCR for GDF15 mRNA. The vertical axis indicates the fold increments compared to the data of the samples at Day 3. Data are shown as mean ± SD (*n* = 3). (**f**) Immunostaining studies for GDF15 protein (green) using the terminally differentiated samples at Day 10. Nuclei were counterstained by DAPI (blue).

**Figure 3 cells-09-01365-f003:**
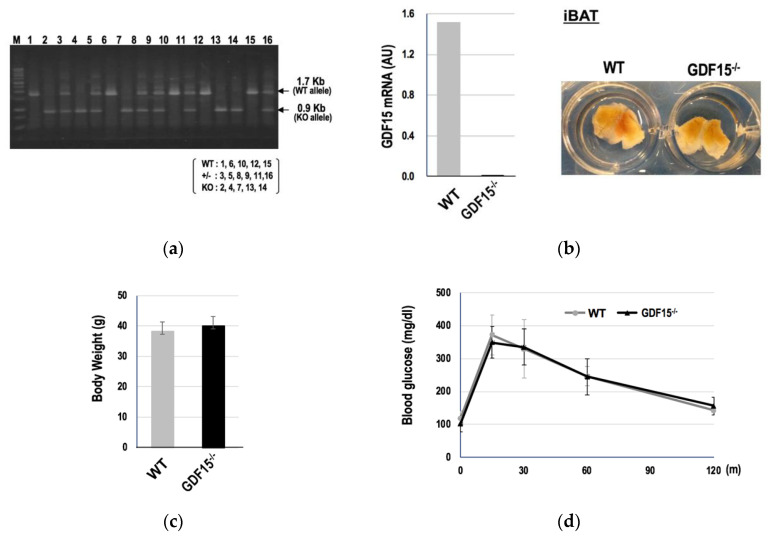
*GDF15^-^*^/-^ mice have normal brown adipose tissue (BAT) and are metabolically healthy. Results of the PCR-based genotyping of the offspring (**a**). *GDF15* mRNA expressions (**b**, **left**) and macroscopic features (**b**, **right**) of the iBAT of wild-type (WT) or *GDF15*^-/-^ mice are shown. There were no significant differences in body weight (**c**) or glucose tolerance, which was evaluated by an oral glucose tolerance test (OGTT) (**d**), between WT and *GDF15^-^*^/-^ mice (WT: *n* = 8; *GDF15*^-/-^: *n* = 15) after an 8-week high-fat diet. Data are shown as mean ± SD.

**Figure 4 cells-09-01365-f004:**
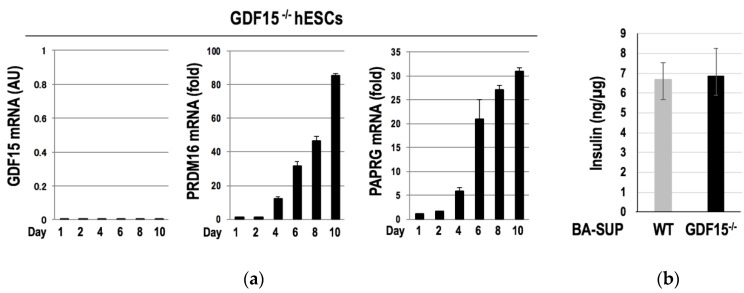
*GDF15^-^*^/-^ hESCs normally differentiated into BAs. *GDF15^-/^*^-^ hESCs were subjected to differentiation into BAs. The expressions of the two major BA-selective genes, *PRDM16* and *PPARG*, were properly induced in *GDF15*^-/-^ hESCs despite the lack of GDF15 expression (*n* = 3 experiments) (**a**). There were no differences in the activities of the BA-SUP to enhance insulin secretion from murine pancreatic beta cells between WT and *GDF15*^-/-^ hESCs (*n* = 3 experiments) (**b**). Data are shown as mean ± SD.

**Figure 5 cells-09-01365-f005:**
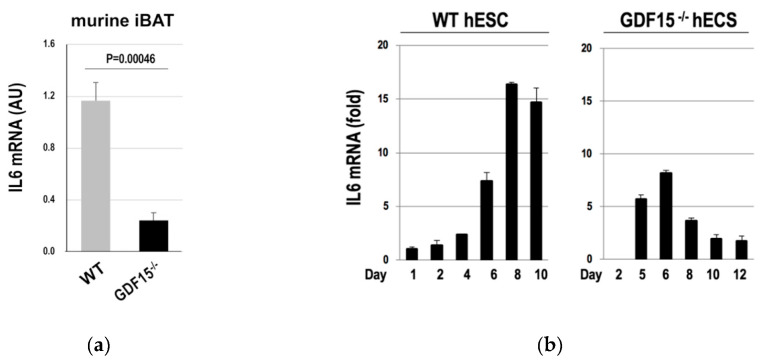
*IL6* expressions were reduced by the absence of *GDF15*. The expression of *IL6* mRNA was measured in the iBATs of WT and *GDF15*^-/-^ mice (*n* = 3 mice) (**a**) or in the hESCs that were subjected to the differentiation into BAs (*n* = 3 experiments) (**b**). Data are shown as mean ± SD.

**Figure 6 cells-09-01365-f006:**
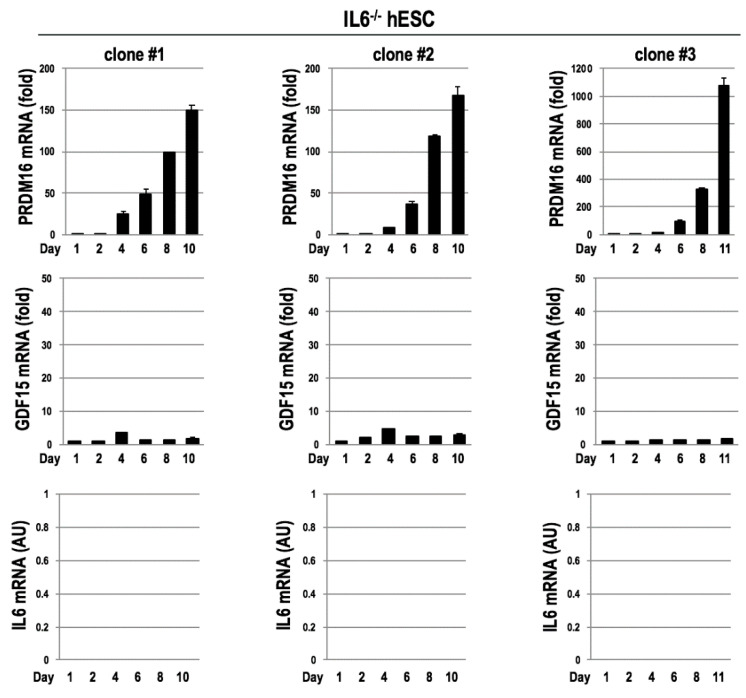
*GDF15* expressions were blunted by the absence of *IL6.* Using three clones of *IL6^-^*^/-^ hESCs, *PRDM16* and *GDF15* expressions were examined in the course of differentiation into BAs. Despite the induction of *PRDM16* (**upper panels**), *GDF15* mRNA was markedly repressed (**middle panels**) (*n* = 3 experiments) by the absence of *IL6* (**lower panels**). Data are shown as mean ± SD.

**Figure 7 cells-09-01365-f007:**
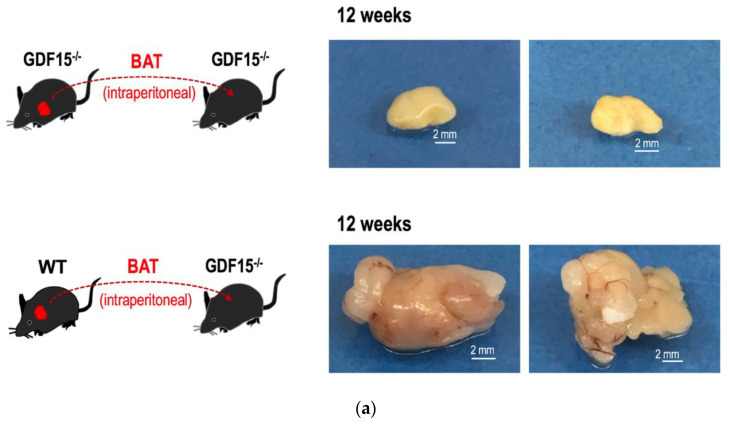
GDF15 is required for long-term survival of BAT grafts. The iBATs of either *GDF15*^-/-^ or WT mice were intraperitoneally transplanted into *GDF15^-^*^/-^ mice according to the method presented by Stanford et al. [20]. After 12 weeks, the grafts were removed. (**a**) Macroscopic observations. The BAT grafts of *GDF15*^-/-^ mice were shrunk and no angiogenesis was detected (**upper**), whereas those of WT mice retained their sizes and neo-angiogenesis was detected in the soft tissue around the grafts (**lower**). (**b**) Histological observations. Tissue slices of BAT grafts derived from *GDF15^-^*^/-^ mice (**left**) and those of WT mice (**right**) were subjected to HE staining. Structures of the elastic membranes of the arteries were well preserved even in *GDF15*^-/-^-derived BAT grafts (**left**, arrows).

**Figure 8 cells-09-01365-f008:**
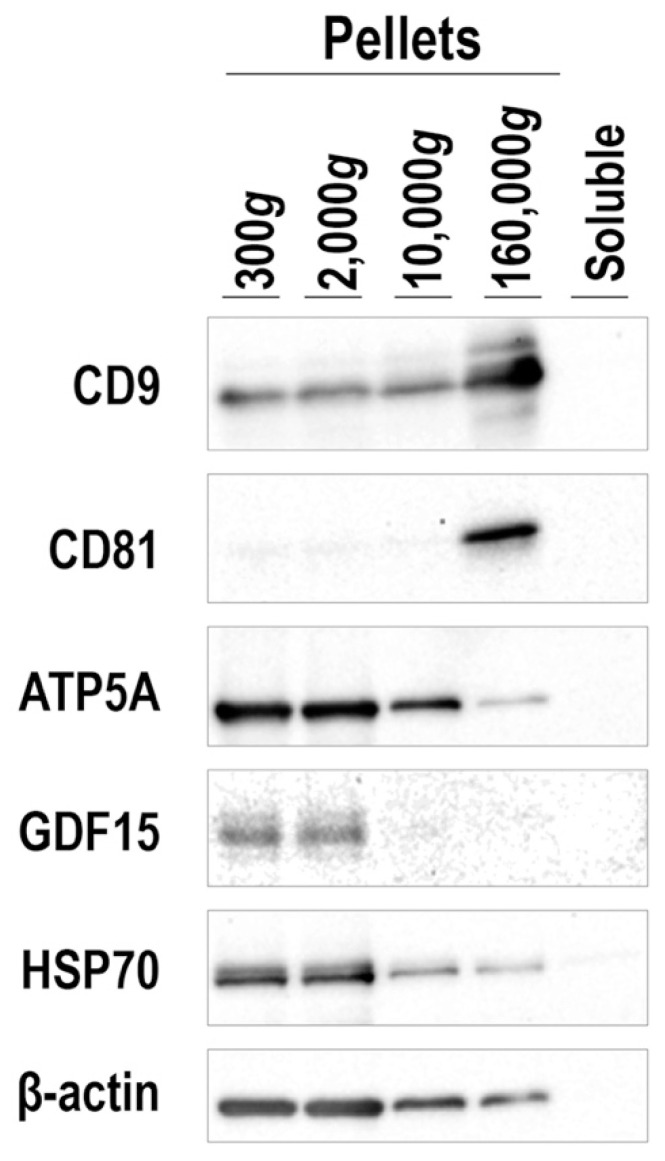
Culture supernatant of hESC-derived BAs was subjected to sequential centrifugation with stepwise increasing gravities as indicated. Western blotting was performed by applying aliquots of lysed pellets obtained in each step using indicated antibodies. “Soluble” indicates the supernatant after centrifugation at 160,000× *g.*

**Figure 9 cells-09-01365-f009:**
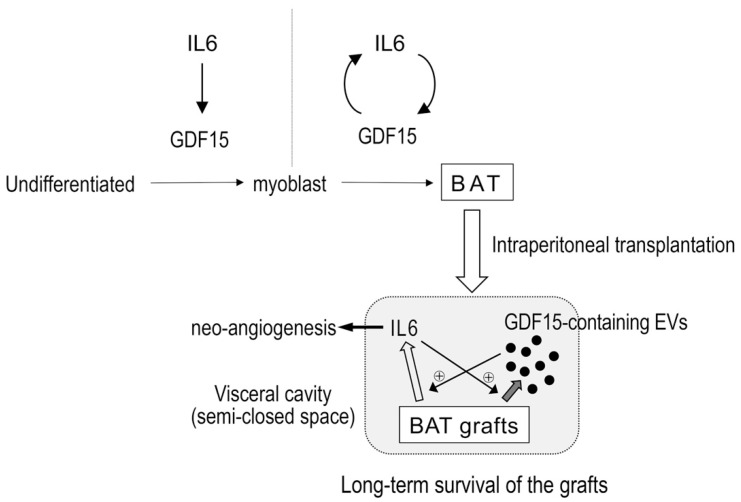
Model. IL6 induces GDF15 in the former phase of the differentiation of hESCs into BAs, whereas GDF15 and IL6 create a mutually inducing loop in the later phase. The GDF15, which may be secreted as a component of large-sized EVs, and IL6 potentiate and stabilize neovascularization, guaranteeing long-term survival of BAT grafts.

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
