# Peer review of "New Role for Growth/Differentiation Factor 15 in the Survival of Transplanted Brown Adipose Tissues in Cooperation with Interleukin-6"

_cells, 2020, doi:10.3390/cells9061365_

Round 1

Reviewer 1 Report

The author have improved the manuscript and clarified missing points. I recommend the manuscript for publication in the present form 

Author Response

We are grateful for valuable comments that the reviewer gave us in our first submission.

Reviewer 2 Report

The Authors describe the update of an interesting approach to obtain brown adipocytes from stem cells, without gene transfer and synthetic cytokine use. They highlight the existence of a mutual loop between GDF15 and IL-6, which is able to promote long-term survival of BAT grafts.

The data are potentially interesting. The paper is rather clearly written.

The methods appear consistent. The results may be a little more concisely written.

Obesity is definitely associated to greater CVD and other diseases risk.

Define better the Obesity Paradox: class I obese subjects with heart failure or CHD may behave better probably because of the relatively greater lean mass than normal or underweight patients. So it is not just a matter of BMI or adipose tissue, but also of lean mass and related issues.

Specific comments

Is the effect of GDF15 on IL-6 expression direct? Or are there any intermediate steps?

What about viceversa?

Please discuss the potential molecular interactions involved in the regulation of the expression of these two molecules.

Some revision of the English language is suggested.

Several typos are present.

Author Response

<The comments by Reviewer 2>

The Authors describe the update of an interesting approach to obtain brown adipocytes from stem cells, without gene transfer and synthetic cytokine use. They highlight the existence of a mutual loop between GDF15 and IL-6, which is able to promote long-term survival of BAT grafts.

The data are potentially interesting. The paper is rather clearly written.

The methods appear consistent. The results may be a little more concisely written.

Obesity is definitely associated to greater CVD and other diseases risk. Define better the Obesity Paradox: class I obese subjects with heart failure or CHD may behave better probably because of the relatively greater lean mass than normal or underweight patients. So it is not just a matter of BMI or adipose tissue, but also of lean mass and related issues.

Specific comments:

Is the effect of GDF15 on IL-6 expression direct? Or are there any intermediate steps? What about viceversa? Please discuss the potential molecular interactions involved in the regulation of the expression of these two molecules.

Some revision of the English language is suggested. Several typos are present.

Point-by-point responses to the comments by reviewer 2:

1) Comment 1: Define better the Obesity Paradox: class I obese subjects with heart failure or CHD may behave better probably because of the relatively greater lean mass than normal or underweight patients. So it is not just a matter of BMI or adipose tissue, but also of lean mass and related issues.

Response 1: According to the valuable comment by the reviewer, we added the following sentences in Introduction. “Overweight/obesity imposes high exercise load on muscles and bones in daily life. Accordingly, overweight and class I obese patients, as long as they are aiming to get adequate exercise in their everyday lives, may behave better owing to the relatively greater lean mass than normal or underweight patients.”

2) Comment 2: Is the effect of GDF15 on IL-6 expression direct? Or are there any intermediate steps? What about viceversa? Please discuss the potential molecular interactions involved in the regulation of the expression of these two molecules.

Response 2: According to the valuable comment by the reviewer, we added the following descriptions in Discussion. “Whether the effect of GDF15 on IL-6 expression and that of IL6 on GDF15 expression are direct or indirect remains elusive. Genomic information from UCSC Genome Browser (https://genome.ucsc.edu/) indicates that there is a relatively large (~ 300 bp) STAT3-binding area within the enhancer region of GDF15 gene as shown by transcription factor CHIP-seq analyses, while there are no SMAD-binding sites shown within the enhancer region of IL6 gene. Therefore, IL6 may possibly induce GDF15 gene expression directly via STAT3 activation while GDF15 acts indirectly to induce IL6 gene. Further investigations are required for the elucidation of the molecular mechanism of mutual gene induction between IL6 and GDF15.”

Below, we show genomic data of GDF15 and IL6 enhancer regions from UCSC Genome Browser for your information.

・Genomic data of GDF15 enhancer regions (please see attached file)

・Genomic data of IL6 enhancer regions (please see attached file)

3) Comment 3: Some revision of the English language is suggested. Several typos are present

Response 3: We have re-checked the English language and corrected typing errors.

Reviewer 3 Report

Through microarray analyses, the authors report that the expression of GDF15, a component of the TGF beta super family, is increased during early differentiation of human embryonic stem cells into brown adipocytes. Because GDF15 is involved in the regulation of metabolism, these authors examined the expression of GDF15 throughout differentiation and found that its expression is linked to IL6. These authors also generated mutant mice lacking GDF15. Surprisingly, these GDF15-/- mice did not have any metabolic phenotypes under the authors’ experimental setup. However, these authors did find that the expression of GDF15 is required for the survival of transplanted BAT in the recipient mice.

This is a revised manuscript from these authors. Additional data, including histological and marker analyses (Ucp1 staining) of the transplanted BAT from wildtype and GDF15-/- mice were provided this time. Also, the authors provide data to suggest that GDF15 can be secreted from brown adipocytes that were differentiated from human embryonic stem cells. Overall, the quality of the manuscript has improved since the last submission. Although the manuscript did not provide mechanistic data supporting the direct link between GDF15 and IL6 or directly test the functional role of GDF15 in the survival of transplanted BAT, the authors did uncover an interesting observation linking GDF15 to the survival of transplanted BAT. However, the manuscript still requires some work, specifically on the data presentation and through providing additional evidence to confirm the authors’ claims regarding the GDF15 transplanted BAT. Following are my comments.

  1. Results 3.1.1. please provide a reference for the Yamanaka factor. Same paragraph, please italicize “NODAL”.
  2. Results 3.1.2. please provide a reference for the statement, GDF15 belongs to TGFB/BMP/GDF family, ------
  3. Some of the qPCR results were presented as fold changes, and some were presented as arbitrary units. Please provide in the methods section how were these done.
  4. Cannot find Fig 6 or any reference to it in the manuscript? Please make sure all the figures are numbered correctly.
  5. Please provide statistical analyses for the gene expression studies.
  6. Results 3.1.4. authors state that “IL6 is indispensable for inducing GDF15 in the differentiating hESCs into BA” based on gene expression data, which did not indicate whether IL6 can directly induce GDF15 expression. Please revise this statement.
  7. How soon dose the transplanted BAT start to degenerate? Additionally, the authors use only GDF15-/- recipient mice for the transplantation. What happens if wildtype recipient mice are used? Would the use of wildtype slow down degeneration of the GDF15-/- transplanted BAT?

Author Response

<The comments by Reviewer 3>

Through microarray analyses, the authors report that the expression of GDF15, a component of the TGF beta super family, is increased during early differentiation of human embryonic stem cells into brown adipocytes. Because GDF15 is involved in the regulation of metabolism, these authors examined the expression of GDF15 throughout differentiation and found that its expression is linked to IL6. These authors also generated mutant mice lacking GDF15. Surprisingly, these GDF15 mice did not have any metabolic phenotypes under the authors’ experimental setup. However, these authors did find that the expression of GDF15 is required for the survival of transplanted BAT in the recipient mice.

This is a revised manuscript from these authors. Additional data, including histological and marker analyses (Ucp1 staining) of the transplanted BAT from wildtype and GDF15 mice were provided this time. Also, the authors provide data to suggest that GDF15 can be secreted from brown adipocytes that were differentiated from human embryonic stem cells. Overall, the quality of the manuscript has improved since the last submission. Although the manuscript did not provide mechanistic data supporting the direct link between GDF15 and IL6 or directly test the functional role of GDF15 in the survival of transplanted BAT, the authors did uncover an interesting observation linking GDF15 to the survival of transplanted BAT. However, the manuscript still requires some work, specifically on the data presentation and through providing additional evidence to confirm the authors’ claims regarding the GDF15 transplanted BAT.

Following are my comments.

  1. Results 3.1.1. please provide a reference for the Yamanaka factor. Same paragraph, please

italicize “NODAL”.

  1. Results 3.1.2. please provide a reference for the statement, GDF15 belongs to TGFB/BMP/GDF family,
  2. Some of the qPCR results were presented as fold changes, and some were presented as

arbitrary units. Please provide in the methods section how were these done.

  1. Cannot find Fig 6 or any reference to it in the manuscript? Please make sure all the figures are numbered correctly.
  2. Please provide statistical analyses for the gene expression studies.
  3. Results 3.1.4. authors state that “IL6 is indispensable for inducing GDF15 in the differentiating hESCs into BA” based on gene expression data, which did not indicate whether IL6 can directly induce GDF15 expression. Please revise this statement.
  4. How soon dose the transplanted BAT start to degenerate? Additionally, the authors use only GDF15 recipient mice for the transplantation. What happens if wildtype recipient mice are used? Would the use of wildtype slow down degeneration of the GDF15 transplanted BAT?

Point-by-point responses to the comments by reviewer 3:

1) Comment 1: Results 3.1.1. please provide a reference for the Yamanaka factor. Same paragraph, please italicize “NODAL”

Response 1: We have provided a reference for the Yamanaka factor (ref 29) and italicized the gene name “NODAL”

2) Comment 2: Results 3.1.2. please provide a reference for the statement, GDF15 belongs to TGFB/BMP/GDF family.

Response 2: We have provided a reference for the statement, GDF15 belongs to TGFB/BMP/GDF family (ref 33). We also corrected the phrase “GDF15 belongs to a TGFB/BMP/GDF family” as “GDF15 belongs to a BMP/GDF family of the TGFB superfamily [33]”. 

3) Comment 3: Some of the qPCR results were presented as fold changes, and some were presented as arbitrary units. Please provide in the methods section how were these done

Response 3: In Materials and Methods (Section 2.3), we described that the results for each mRNA level were normalized against those for GAPDH.

4) Comment 4: Cannot find Fig 6 or any reference to it in the manuscript? Please make sure all the figures are numbered correctly.

Response 4: We deeply apologize for our mistake. We have checked the number of all figures and corrected the errors.

5) Comment 5: Please provide statistical analyses for the gene expression studies

Response 5: We described the method of statistical analyses in Materials and Methods (Section 2.7) and the figure legends.    

6) Comment 6: Results 3.1.4. authors state that “IL6 is indispensable for inducing GDF15 in the differentiating hESCs into BA” based on gene expression data, which did not indicate whether IL6 can directly induce GDF15 expression. Please revise this statement

Response 6: We appreciate your valuable comment. We changed the phrase “indicating that IL6 is indispensable for inducing GDF15 in the differentiating hESCs into BA. indicating that IL6” as “indicating that IL6 plays an important role in inducing GDF15 in the differentiating hESCs into BA”.

7) Comment 7: How soon dose the transplanted BAT start to degenerate? Additionally, the authors use only GDF15 recipient mice for the transplantation. What happens if wildtype recipient mice are used? Would the use of wildtype slow down degeneration of the GDF15 transplanted BAT?

Response 7: We appreciate your valuable comment. We added descriptions regarding this point in Discussion as follows. 

“Although it remains elusive when massive cell death in GDF15-/- mice-derived BAT grafts started, it seems that the commitment of cell death occurred at an early phase after transplantation. We had observed in our previous experiments of organ culture of iBAT that the morphology of BA turned to a white adipocyte-like feature, with a large monolocular lipid droplet, within 24 hours (unpublished observations). On the other hand, dead BAs in GDF15-/- mice-derived BAT grafts still bore multilocular lipid droplets. Therefore, signals to induce cell death should have been transmitted within 24 hours after transplantation. Future investigations may clarify the detailed process of the unique cell death executed in ischemic BAT.”   

Round 2

Reviewer 2 Report

All reviwer's comments have been properly addressed.

Reviewer 3 Report

These authors addressed most of my comments. I have no additional comments for them.

This manuscript is a resubmission of an earlier submission. The following is a list of the peer review reports and author responses from that submission.

Round 1

Reviewer 1 Report

Oka M et al. claim that there is a GDF15-IL6 cooperation which facilitates the survival of transplanted BAT. Overall I am not convinced that that experiments presented are the appropriate to support the conclusions. I find the explanation of linking GDF15-Il6 in the context of BAT transplantation survival odd and not really substantiated ( the authors say that they choose to look at IL6 because it shows the same tissue expression profile as GDF15- could be any other cytokine also ? ) . Also I find the presentation of data poor and not clear. In vivo experiments need further and complete phenotypic characterization to support any claims made here. More of my specific comments are below. My general recommendation is against publication. 

Line 200-202: Are LEFTY1, LEFTY2, NODAL and CITED1 established gene  markers of commitement towards BA differentiation ? Is there any knowledge whether deletion or overexpression of those genes affects the differentiation of hESCs into BA? Please add references. 

Figure 1 and SupplFig1.: Please add units and description of X and Y axis. Also statistics should be performed and information on what is measured should be included in the figure legend. 

Brown adipose tissue is indicated as iBAT , BAT or BA. Please keep one abbreviation consistent throughout the paper. 

It is odd that GDF15 deletion gives no phenotype uponHFD.Previously another mouse GDF15KO mouse  (ref 32 at the paper) has been published in similar mouse genetic background as the one use in the present manuscript and gives a strong glucose intolerant phenotype upon HFD) Since the authors are using a Crispr Cas 9 mouse which is another method that the one used before /see ref. 32) they should measure deletion of GDF15 in other tissue next to the BAT eg. liver (since OGTT would mainly represent liver insulin resistance).Also food intake data and body composition should be shown. Importantly circulating GDF15 levels should be shown to confirm complete deletion of GDF15 from the system in the KO mice. 

line 348-349: This can be simply due to the effects of GDF15 in wound healing  ,survival and angiogenesis. Thus, GDF15 may not be necessarily required for the survival of BAT grafts, but for the survival of grafts of any tissue origin. Are IL6 circulating levels different between the WT and GDF15 KO mice ? To make the link to the rest of the vitro data circulating levels of IL6 should be measure in the GDF15KO mice. 

Reviewer 2 Report

The paper by Oka and Coauthors reports an interesting study on the cooperative interaction between GDF15 and IL6 to promote the survival of transplanted brown adipose tissue.

The main data are convincing and the hypothesis that has been put forward seems to be well supported by the findings obtained.

The paper is clearly written and well understandable.

The text requires however an extensive revision of the English language and the correction of numerous typos (eg, lines 187 to 191, etc.).

Specific comments

In the title, IL6 may be better indicated in full as “interleukin-6”.

Introduction: the concept and potential usefulness of BAT transplantation for morbid obesity should be better explained.

M&M and Results section appear complete and informative.

Discussion: this section is largely devoted to describe GDF15 biology and actions, instead of discussing  the relevance of the obtained findings, especially in the context of the potential hESC/hiPSC-derived BAT cell transplantation in obesity.

Thus, GDF15 information appear to be in the wrong place and may in part be included in the Introduction. In any case, the Discussion needs to be rewritten with a stronger focus on the relevance of these findings.

Figure 9: Correct the term “Neovasulization” to “Neovascularization”

Reviewer 3 Report

In this manuscript, the authors provide data that suggest GDF15, through coordination with IL6, is necessary for the survival of transplanted brown adipose tissues. This work is basically an extension from the authors’ previous studies using human embryonic stem cells to generate brown adipocytes with specific induction medium. During the course of differentiation of these stem cells, these authors found that GDF15 expression increased. Thus, they investigate the role of GDF15 in brown adipose tissue development in vitro and in vivo. However, despite the increase of GDF15 expression level, loss of GDF15 does not impact brown adipose tissue development and function. Interestingly, these authors found that GDF15-/- brown adipose transplanted degenerated, while wildtype brown adipose tissue did not. Overall, this manuscript provides interesting observations but lacks needed in-depth analyses, specifically for the GDF15-/- brown adipose tissue transplantation experiment. In addition, the manuscript text, and figure presentation need major work to improve its clarity. Thus, this manuscript is not suitable for publication at this current form. Following are the review comments.

More analyses are needed for the GDF15/- brown adipose tissue transplantation studies as they relate to the most important point of this manuscript. For example, histological and marker analyses of the transplanted tissue are needed because they may provide some insight regarding the cause of transplanted brown adipose tissue degradation. It is not clear when the degradation started. Additionally, the authors only use only GDF15-/- recipient mice for the transplantation. What happens if the wildtype recipient mice are used? Cannot find Fig 6 or any reference to it in the manuscript? Please make sure all the figures are numbered correctly. Please provide statistical analyses to the gene expression studies. Please provide titles for the X and Y axes in figures. Some are presented and not are not. Include background info regarding GDF15 in the introduction because it is the main gene studied in here.